# **Evaluating the performance of CMIP6 models in simulating Southern Ocean biogeochemistry**

Ming Cheng<sup>1,2</sup>, Nicola Maher<sup>3</sup>, Michael J. Ellwood<sup>1,2</sup>

Abstract. The Southern Ocean plays a vital role in global biogeochemical cycles, yet comprehensive assessments of its representation in Earth System Models (ESMs) are still limited. This study evaluates the performance of 14 Coupled Model Intercomparison Project Phase 6 (CMIP6) models in simulating key biogeochemical variables south of 30°S, including austral-summer surface chlorophyll, deep chlorophyll maxima (DCMs), nitrate, silicate, dissolved iron, and particulate organic carbon (POC). Model output for the period 2000-2014 is compared to multiple observational datasets, such as a Copernicus product for estimated chlorophyll and POC profiles, the World Ocean Atlas (WOA) for nitrate and silicate, and GEOTRACES products for dissolved iron. Model performance is assessed using statistical metrics including mean bias error (MBE), standardised standard deviation (SSD), root mean squared deviation (RMSD), and correlation coefficient (CC). The results reveal substantial inter-model variability, with individual models exhibiting strengths in simulating different variables. GFDL-ESM4 best reproduces surface chlorophyll and POC and DCM patterns, and IPSL-CM6A-LR performs best for all nutrients, including nitrate, silicate, and dissolved iron. Based on composite rankings, the top-performing models are IPSL-CM6A-LR, GFDL-ESM4, CNRM-ESM2-1, UKESM1-0-LL, and CMCC-ESM2. This work underscores the importance of multi-model evaluation for identifying model strengths and guiding future improvements in biogeochemical (BGC) model development, particularly in the context of understanding and projecting Southern Ocean biogeochemistry under climate change.

# 1 Introduction

10

Climate change is a critical global challenge, driving major shifts in marine conditions and ecosystems. The Southern Ocean, covering 30% of the global ocean, plays a crucial role in the oceanic carbon and nutrient cycles, absorbing over 40% of anthropogenic CO<sub>2</sub> and 70% of human-induced warming (Gruber et al., 2019; Petrou et al., 2016; Xue et al., 2024). The Southern Ocean is characterised by complex interactions among physical circulation, biogeochemistry, and biological productivity, making it a challenge to model (Henley et al., 2020; Morley et al., 2020). The powerful eastward-moving

<sup>&</sup>lt;sup>1</sup>Research School of Earth Sciences, Australian National University, Canberra, ACT 2601, Australia

<sup>&</sup>lt;sup>2</sup>Australian Centre for Excellence in Antarctic Science, Research School of Earth Sciences, Australian National University, Canberra, ACT 2601, Australia

<sup>&</sup>lt;sup>3</sup>ARC Centre of Excellence for Weather of the 21<sup>st</sup> Century, Australian National University, Canberra, ACT 2601, Australia *Correspondence to*: Ming Cheng (ming.cheng@anu.edu.au)

Antarctic Circumpolar Current (ACC), one of the Earth's strongest currents, connects ocean basins and regulates global climate and ocean circulation, supports diverse marine ecosystems, and distributes nutrients (Böning et al., 2008; Rintoul et al., 2001; Lopes et al., 2011; Song, 2020). The upwelling of deep, nutrient-rich waters, driven by ACC, supports phytoplankton growth, influencing global carbon sequestration and ecosystem dynamics (Venables and Moore, 2010; Morrison et al., 2015; Hunt et al., 2021; Pollard et al., 2006). This complex region of both physical and biological processes is important due to its significant impact on global climate regulation, carbon sequestration, and the health of marine ecosystems.

Phytoplankton, particularly silicifying diatoms, are a key component of the Southern Ocean food web and the global carbon cycle, playing a crucial role in carbon sequestration and nutrient cycling (Deppeler and Davidson, 2017; Baldry et al., 2020; Petrou et al., 2016; Nissen and Vogt, 2021; Timmermans et al., 2004; Hoffmann et al., 2008). Their biomass and primary production are often assessed through chlorophyll concentrations, which serve as an essential indicator in oceanic carbon fixation and ecosystem productivity (Carranza and Gille, 2015; Johnson et al., 2013). However, despite the abundance of macronutrients such as nitrate and silicate, phytoplankton growth is frequently constrained by light limitation and iron deficiency, both of which regulate their distribution and productivity (Boyd and Ellwood, 2010; Boyd, 2002). In response to these physiochemical conditions, deep chlorophyll maxima (DCMs) have been observed in nutrient-stratified waters during austral summer in the Southern Ocean, indicating robust phytoplankton production in the subsurface layer (Boyd et al., 2024; Cornec et al., 2021; Cullen, 1982; Cullen, 2015; Hopkinson and Barbeau, 2008; Li et al., 2012). These DCMs contribute significantly to the regional carbon cycle, for example, approximately 40% of primary production in the Southern Ocean occurs below the mixed layer (Vives et al., 2024), and support marine food webs by sustaining primary production below the surface, where light and nutrient conditions are more favourable, particularly the supply of iron and silicon, for certain phytoplankton communities (Signorini et al., 2015; Cornec et al., 2021; Sauzède et al., 2018).




Ocean biogeochemical (BGC) modules, are an important component of coupled Earth system models (ESMs), and are indispensable for understanding the complicated physical and biogeochemical processes in the ocean (Follows and Dutkiewicz, 2011; Séférian et al., 2020). Depending on their complexity, these models simulate the cycles of key elements such as carbon, oxygen, nitrogen, phosphorus, silicate, and iron, and organisms including phytoplankton, zooplankton and bacteria, which are vital for marine ecosystems and global climate regulation (Dunne et al., 2020; Aumont et al., 2015; Pak et al., 2021; Ilyina et al., 2013). BGC models enable researchers to investigate how changes in environmental conditions, such as temperature, light, and nutrient availability, impact marine biogeochemistry and ecosystem dynamics (Kwiatkowski et al., 2020). They are particularly valuable for studying regions like the Southern Ocean, where observational data are limited, and the interactions between physical and biogeochemical processes are highly complex (Tagliabue et al., 2017; Lauderdale et al., 2017). Despite their significance, BGC models face considerable challenges, including the need for precise parameterisation of key biological processes, accurate representation of small-scale processes, and effective constrained by diverse data sources (Ackermann et al., 2024; Beadling et al., 2019).

The Coupled Model Intercomparison Project Phase 6 (CMIP6) represents the latest advancement in climate modelling, providing a standardised framework for evaluating ESMs across various simulations under different climate scenarios (Eyring et al., 2016; O'Neill et al., 2016; Meehl et al., 2020). Compared to previous phases, CMIP6 models feature higher spatial resolution, improved physical processes, and enhanced biogeochemical components, including expanded phytoplankton functional types, refined biogeochemical cycle representations and optimised parameterisation (Séférian et al., 2020; Kwiatkowski et al., 2020). However, significant discrepancies persist in biogeochemical performance due to variations in BGC model structures, parameterisation, and ocean physics (Séférian et al., 2020). Evaluating CMIP6 models highlights these differences, offering insights for future model development and refinement (Kwiatkowski et al., 2020; Séférian et al., 2020; Hauck et al., 2015).

While some studies have assessed the performance of CMIP6 models in simulating biogeochemical variables globally and regionally, a comprehensive analysis of chlorophyll, nutrient distribution, and DCM characteristics in the Southern Ocean remains unexplored. Marshal et al. (2024) evaluated chlorophyll, phytoplankton, nitrate and dissolved oxygen across 13 CMIP6 models in the South China Sea, ranking them using statistical metrics to identify the five best-performing models. Fisher et al. (2025) synthesised CMIP6 outputs to examine climate-driven shifts in Southern Ocean primary production, projecting a 30% increase in Antarctic zone productivity under a high-emission (SSP5-8.5) scenario, albeit with regional variations. Séférian et al. (2020) compared CMIP5 and CMIP6 models, demonstrating improved CMIP6 biogeochemical representations, including chlorophyll, dissolved oxygen, silicate and nitrate, due to more comprehensive biogeochemical cycles and Earth system interactions. Rohr et al. (2023) analysed 11 CMIP6 models and found that zooplankton grazing parameterisation introduced uncertainty in marine carbon cycle projections. These studies underscore the need for further evaluation of the CMIP6 models to assess the impact of biogeochemical processes and parameterisation on model performance.

In this paper, we evaluate biogeochemical variables, including chlorophyll, silicate, nitrate, dissolved iron, and particulate organic carbon (POC) across 14 CMIP6 models and assess their performance in representing DCMs in the Southern Ocean. Section 2 details the observed and simulated data and the statistical analysis methods. Section 3 presents an inter-model evaluation of each biogeochemical variable. Section 4 discusses the ocean vertical carbon structure, model performance, and avenues for improvement. Section 5 provides a summary of our findings.

# 2 Data and methods

# 90 2.1 Study region




This study focuses on the open waters of the Southern Ocean (south of 30°S). We divide the Southern Ocean into four zones: the subtropical zone (STZ), subantarctic zone (SAZ), polar front zone (PFZ) and Antarctic zone (AZ; Fig. 1). These zones are separated by three key fronts: the subtropical front, subantarctic front and polar front, which are defined by distinct

physical and biogeochemical properties (Orsi et al., 1995). We compare the CMIP6 model outputs of chlorophyll, nitrate, silicate, dissolved iron, and POC across these zones and across the entire Southern Ocean.

# 2.2 CMIP6 datasets and availability




We obtained outputs from 14 CMIP6 models from the Earth System Grid Federation (ESGF) Nodes (Cinquini et al., 2014). Specifically, we collected data from the historical experiment for model evaluation, using the ensemble member r1i1p1f1 for most models, while r1i1p1f2 was used for CNRM-ESM2-1, MIROC-ES2L, and UKESM1-0-LL. Dissolved iron and carbon data of ACCESS-ESM1-5 were collected from the National Computational Infrastructure. This includes monthly data for chlorophyll, nitrate, silicate, dissolved iron, POC which comprises phytoplankton, detritus, and bacteria (see Section 2.4 for details), and NPP. These data were also used to compare chlorophyll and DCM distribution. The selected CMIP6 models, their properties and available variables are detailed in Table 1. To ensure consistency, we regridded all outputs to a 1°×1° common horizontal resolution using bilinear interpolation in Climate Data Operators (CDO) software (Schulzweida, 2023), covering the time range from January 2000 to December 2014.

Table 1. List of 14 CMIP6 models utilised, detailing the ESM name, coupled ocean biogeochemical model (OBGCM) name, averaged horizontal resolution and variables with available data. All variable abbreviations and their long names: chl (mass concentration of phytoplankton expressed as chlorophyll in seawater), no3 (dissolved nitrate concentration), si (total dissolved inorganic silicon concentration), dfe (dissolved iron concentration), phyc (phytoplankton carbon concentration), detoc (mole concentration of organic detritus expressed as carbon in seawater), bacc (bacterial carbon concentration), and intpp (integrated net primary production).

| ESM             | OBGCM         | Variable                                    | ESM and OBGCM Reference                       |
|-----------------|---------------|---------------------------------------------|-----------------------------------------------|
| ACCESS-ESM1-5   | WOMBAT        | chl, no3, dfe, phyc, detoc, intpp           | Ziehn et al. (2020); Oke et al. (2013)        |
| CanESM5         | CMOC          | chl, no3, phyc, detoc                       | Swart et al. (2019); Zahariev et al. (2007)   |
| CESM2           | MARBL         | chl, no3, si, dfe, phyc, intpp              | Danabasoglu et al. (2020); Long et al. (2021) |
| CMCC-ESM2       | BFM v5.2      | chl, no3, si, dfe, phyc, detoc, bacc, intpp | Lovato et al. (2022); Vichi et al. (2015)     |
| CNRM-ESM2-1     | PISCES-v2-gas | chl, no3, si, dfe, phyc, detoc, intpp       | Séférian et al. (2019); Skyllas (2018)        |
| GFDL-ESM4       | COLBALTv2     | chl, no3, si, dfe, phyc, detoc, bacc, intpp | Dunne et al. (2020); Stock et al. (2020)      |
| IPSL-CM6A-LR    | PISCES-v2     | chl, no3, si, dfe, phyc, detoc, intpp       | Boucher et al. (2020); Aumont et al. (2015)   |
| MIROC-ES2L      | OECO-v2       | chl, no3, dfe, phyc, intpp                  | Hajima et al. (2020)                          |
| MPI-ESM-1-2-HAM | HAMOCC6       | chl, no3, si, dfe, phyc, detoc, intpp       | Neubauer et al. (2019); Ilyina et al. (2013)  |
| MPI-ESM1-2-HR   | HAMOCC6       | chl, no3, si, dfe, phyc, detoc, intpp       | Müller et al. (2018); Ilyina et al. (2013)    |
| MPI-ESM1-2-LR   | HAMOCC6       | chl, no3, si, dfe, phyc, detoc, intpp       | Mauritsen et al. (2019); Ilyina et al. (2013) |
| NorESM2-LM      | HAMOCC        | chl, no3, si, dfe, phyc, detoc, intpp       | Tjiputra et al. (2020)                        |
| NorESM2-MM      | HAMOCC        | chl, no3, si, dfe, phyc, detoc, intpp       | Tjiputra et al. (2020)                        |
| UKESM1-0-LL     | MEDUSA-2.0    | chl, no3, si, dfe, phyc, detoc, intpp       | Sellar et al. (2019); Yool et al. (2013)      |

#### 2.3 Observed datasets and availability







Observed surface chlorophyll and POC data, as well as vertical chlorophyll profiles, were obtained from the Copernicus Global Ocean 3D Chlorophyll-a Concentration, Particulate Backscattering coefficient and Particulate Organic Carbon product (hereafter referred to as the Copernicus product; Sauzède et al., 2016). The original POC fields were estimated using a neural network approach (machine-learning method) known as SOCA2016 (Satellite Ocean Colour merged with Argo data; 2016 version), which integrates satellite-derived surface estimates of the particulate backscattering coefficient (bbp) and chlorophyll a concentration with depth-resolved physical properties derived from BGC-Argo floats (Sauzède et al., 2016). In this study, we used the updated SOCA2024 product, which merges bbp and chlorophyll-a concentrations derived from both satellite and BGC-Argo floats. This product has been validated against a global independent pigment dataset obtained from High Performance Liquid Chromatography (HPLC) and BGC-Argo floats (Sauzède et al., 2024). The reprocessed dataset provides a spatial resolution of 0.25°×0.25° and 36 vertical levels from the surface to 1000 m depth, with a bathymetric mask applied to exclude grid cells shallower than 1000 m to avoid potential coastal artifacts (Sauzède et al., 2024). This masking, however, removes several productive shelf regions such as around New Zealand and the Patagonian Shelf. The primary advantage of this dataset lies in its integration of satellite and BGC-Argo data, ensuring consistent vertical and horizontal chlorophyll and POC distributions. To verify the reliability of the Copernicus dataset, we compared multiple Copernicus versions and satellite-derived products using different algorithms from Johnson et al. (2013). Chlorophyll estimated with SOCA2024 showed the strongest agreement with BGC-Argo data (R<sup>2</sup>=0.89, slope=1.04; Table S1), outperforming other approaches. When validated against HPLC data, SOCA2024 also performed well (R<sup>2</sup>=0.53, slope=0.70) globally, similar to the MODIS-Aqua chlorophyll recalculated using the new algorithm of Johnson et al. (2013), which remains one of the best-performing satellite products for the Southern Ocean (Table S1). The SOCA2024 POC dataset also proved robust when compared with BGC-Argo bbp data in the Southern Ocean (R<sup>2</sup>=0.93, slope=0.89; Table S1). Moreover, a direct comparison between surface chlorophyll from Copernicus product and MODIS (Johnson et al., 2013) reveals a strong correlation (R<sup>2</sup>=0.69, slope=0.80; Fig. S1), confirming the reliability of the Copernicus dataset. Accordingly, we use the Copernicus chlorophyll and POC products for all subsequent analyses of surface chlorophyll, POC, and DCMs to maintain consistency across different observational comparisons. Additionally, we provide model-data comparison using Aqua-MODIS chlorophyll derived with the Johnson et al. (2013) algorithm in the supplement for cross-validation.

Observed nitrate and silicate data were sourced from the World Ocean Atlas (WOA) 2018 (Garcia et al., 2019), representing climatological averages from 1955 to 2017. Observed dissolved iron data were obtained from GEOTRACES IDP2021v2 and a global compilation of dissolved iron measurements (Tagliabue et al., 2012), which compiles bottle-sampled dissolved iron measurements from 2001 to 2014.

Existing NPP estimating algorithms exhibit large discrepancies in the Southern Ocean, with values ranging from approximately 400 to 1400 mg C m<sup>-2</sup> d<sup>-1</sup> in December (Silsbe et al., 2016). The scarcity of *in situ* NPP measurements makes it difficult to constrain or validate these NPP algorithms, introducing substantial uncertainty into any evaluation. Thus, this

study does not include NPP in the model performance ranking to avoid the influence of uncertain reference datasets. Instead, we provide model-data comparison figures in the Supplement (Figs. S2-5) for readers interested in model NPP performance, using NPP products derived from the Standard VGPM (Behrenfeld and Falkowski, 1997) and CAFE (Silsbe et al., 2016) algorithms available from the Ocean Productivity site (<a href="https://orca.science.oregonstate.edu/index.php">https://orca.science.oregonstate.edu/index.php</a>, accessed on 3 October 2025). Both algorithms estimate NPP using MODIS satellite inputs, including chlorophyll a concentration, sea surface temperature, and photosynthetically available radiation, among other parameters.

#### 2.4 Data analysis







To evaluate the performance of CMIP6 models in simulating biogeochemical variables, we compared observations with model outputs for chlorophyll, nitrate, silicate and dissolved iron (Section 3.1), DCM (peak of chlorophyll concentration in the subsurface) representation and characteristics (Section 3.2) and particulate organic carbon (Section 3.3) and presented model rankings by variable (Section 3.4).

Since Southern Ocean DCMs predominantly occur during austral summer (Cornec et al., 2021; Prakash and Bhaskar, 2024), all datasets (except observed dissolved iron) were restricted to December, January and February (DJF). We calculated temporal averages for CMIP6-simulated variables and Copernicus chlorophyll and POC profiles over DJF from 2000 to 2014. Similarly, we computed DJF-averaged nitrate and silicate from observations.

The observed dissolved iron data are distributed sporadically because of the limited source of bottled samples and towed fish samples. Tagliabue et al. (2012) compiled the bottled samples from voyages, and GEOTRACES IDP2021v2 have multiple sources of dissolved iron data, including bottled samples and towed fish samples. To match the surface dissolved iron south of  $30^{\circ}$ S in the model dataset, here we selected the observed dissolved iron data points south of  $30^{\circ}$ S and the depths no deeper than 30 m to represent the surface. In this case, there are only 834 and 910 data points in these spatial ranges, in GEOTRACES IDP2021v2 and Tagliabue et al. (2012), respectively. We merged these two datasets by removing the data points with the same longitude, latitude, depth and dissolved iron concentration. There are finally 1693 unduplicated data points. To mitigate the impact of uneven spatial sampling density on the overall assessment, we gridded all observational data points onto a  $1^{\circ} \times 1^{\circ}$  longitude—latitude mesh grid and aggregated multiple observations within the same grid cell (using the median) to obtain a representative value for subsequent analyses. Finally, there are 615 grid points with available dissolved iron concentrations, accounting for 4.33% of the total ocean grid points. In this research, we only compared the surface dissolved iron concentrations where the observed dissolved iron concentrations are available.

In cases where CMIP6 models do not provide a specific variable representing total particulate organic carbon (POC), we manually derive it by summing different species of POC. The simulated POC concentration in this paper is calculated as the sum of phytoplankton carbon, detrital organic carbon (absent in CESM2 and unavailable in MIROC-ES2L), and bacterial carbon (available only in CMCC-ESM2 and GFDL-ESM4).

To quantify model performance, we calculated spatial variations, mean bias error (MBE), standardised standard deviation (SSD), correlation coefficient (CC), and root mean squared deviation (RMSD) for chlorophyll, nitrate, silicate and dissolved iron. We visualised spatial variations using Southern Ocean maps, MBE in bar charts, SSD, CC and RMSD using Taylor Diagram (TD) to illustrate the agreement between models and observations (Taylor, 2001). The TDs and their related statistics-SSD, CC, and RMSD-are provided in Supplementary Materials.

DCMs are identified as the vertical peak of chlorophyll concentration, where the chlorophyll value exceeds 1.1 times the surface chlorophyll concentration. The 1.1 threshold is applied to account for potential measurement errors in the observation. To evaluate DCM characteristics, we calculated peak chlorophyll concentration at the identified DCM depth and frequency of DCM occurrence, which is defined as the area proportion where DCMs are detected.

The assessment of CMIP6 model performance relies on the ranking of four statistical metrics, containing MBE (the lower |MBE| has the higher ranking), SSD (the closer to 1 has the higher ranking), RMSD (the lower has the higher ranking), and CC (the higher has the higher ranking) for chlorophyll, nitrate, silicate, dissolved iron, and POC. For the evaluation of DCMs, both the chlorophyll rank and DCM occurrence frequency (the closer to the reference, the higher ranking) are considered. The ranking score for each variable is calculated by averaging the rankings of its relevant statistical metrics. That means the model with a lower ranking score has a higher rank. The overall ranking score of each CMIP6 model is calculated by averaging the ranking score of all variables, where the model with a lower ranking score has a higher overall rank. We will present the ranking of each variable and the overall ranking in Section 3.4.

All data processing and analysis were performed using MATLAB R2024a and its numerical toolboxes. Maps were generated using the M\_Map toolbox (Pawlowicz, 2020). Taylor diagrams in supplementary materials were generated using MATLAB functions from Haroon Haider (<a href="https://www.youtube.com/@EngrHaroonHaider">https://www.youtube.com/@EngrHaroonHaider</a>, accessed on: 22 April 2025).

#### 3 Results






#### 3.1 Southern Ocean biogeochemistry

We evaluate the performance of 14 CMIP6 models in simulating Southern Ocean biogeochemistry by comparing their outputs for chlorophyll, nitrate, silicate, and dissolved iron with observational data. The surface chlorophyll concentration in the Southern Ocean exhibits a general increase from north to south, reaching its highest concentrations in the coastal regions of Antarctica (Fig. 1), with some exceptions associated with island wake effects related to continental iron input (Blain et al., 2007). Another main exception is the exceptionally high chlorophyll on the Patagonian Shelf off southeastern South America (Fig. S6), driven by the convergence of the nutrient-rich Malvinas Current with the warm Brazil Current, in addition to shelf upwelling and riverine inputs (Piola et al., 2024; Ferreira et al., 2009; Rivas et al., 2006; Rijkenberg et al., 2014). However, none of the CMIP6 models reproduce this feature, likely because their coarse 1° resolution fails to resolve critical shelf-front dynamics.

Figure 1: Observed surface chlorophyll concentrations from Copernicus in DJF and spatial biases of surface chlorophyll concentrations for 14 CMIP6 models (model chlorophyll – Copernicus chlorophyll) in DJF for the Southern Ocean (>30°S). Black dashed lines in the maps denote the subtropical front, the subantarctic front, and the polar front, from north to south. Grey areas denote regions where no data are available.

Most models underestimate the surface chlorophyll south of the subtropical front, except for the three MPI-ESMs, and many models underestimate the surface chlorophyll in the subtropical zone (Fig. 1). This discrepancy potentially reflects methodological differences: models include only chlorophyll in live phytoplankton, whereas satellites detect chlorophyll in both living and senescent cells. A slight overestimation in the Copernicus chlorophyll product (slope=1.04; Table S1; Sauzède et al., 2024) may also contribute. The three MPI-ESMs, MPI-ESM-1-2-HAM, MPI-ESM1-2-HR, and MPI-ESM1-2-LR, tend to substantially overestimate chlorophyll concentrations throughout the Southern Ocean, with MBEs of 1.03, 1.79 and 0.76 mg/m³ (Fig. 2), respectively, compared to a mean chlorophyll concentration of only 0.59 mg m⁻³ in observations. Conversely, the CanESM5, CMCC-ESM2, CNRM-ESM2-1, and IPSL-CM6A-LR models underestimate chlorophyll concentrations (Figs. 1 and 2). The ACCESS-ESM1-5, CESM2, MIROC-ES2L, NorESM2-LM, NorESM2-MM, and UKESM1-0-LL models exhibit small and negative MBEs for the entire Southern Ocean but show opposing biases across regions. For instance, they overestimate chlorophyll concentrations north of the subtropical front and underestimate concentrations to the south (Fig. 2). The GFDL-ESM4 model provides the most realistic simulation of chlorophyll concentration north of the polar front but underestimates concentrations south of the polar front (Fig. 1).

Figure 2: The mean bias errors in surface chlorophyll concentrations for the Southern Ocean (SO), the subtropical zone (STZ), the subantarctic zone (SAZ), the polar front zone (PFZ), and the Antarctic zone (AZ) in DJF. All MBEs and means in each region are calculated using area-weighted averages.

When considering other metrics such as standardised standard deviation (SSD), root mean-squared deviation (RMSD), and correlation coefficient (CC), we find that among the models, GFDL-ESM4, IPSL-CM6A-LR, and CMCC-ESM2 have the lowest RMSD, small bias errors, and CC values above 0.6, indicating that they were the best-performing models for simulating the distribution of chlorophyll across the Southern Ocean (Fig. S7 and Table S2). In contrast, the three MPI-ESMs are less reliable due to their overestimation of chlorophyll concentration. Additionally, the ACCESS-ESM1-5, CanESM5, and NorESMs models exhibit poor performance, such as their low CC (

Figure 3: Observed surface nitrate concentrations from WOA in DJF and spatial biases of surface nitrate concentrations for 14 CMIP6 models (model nitrate – WOA nitrate) in DJF for the Southern Ocean (>30°S).

Figure 4: The mean bias errors in surface nitrate concentrations for the Southern Ocean (SO), the subtropical zone (STZ), the subantarctic zone (SAZ), the polar front zone (PFZ), and the Antarctic zone (AZ) in DJF.

Among the 14 CMIP6 models, IPSL-CM6A-LR, GFDL-ESM4, and CNRM-ESM2-1 produce the most accurate simulations of surface nitrate concentration for the Southern Ocean. They exhibit the lowest RMSD (<0.3), minimal MBE (absolute MBE < 4 mmol/m³), high CC (>0.95), and SSDs close to 1, indicating strong agreement with observations (Fig. S8 and Table S3). Conversely, the three MPI-ESMs models produce less accurate simulations of surface nitrate concentration for the Southern Ocean due to their large bias errors and significant deviations (represented by SSD, RMSD, and CC on a Taylor diagram; Fig. S8 and Table S3). The remaining models including ACCESS-ESM1-5, CanESM5, CESM2, CMCC-ESM2, MIROC-ES2L, NorESM2-LM, NorESM2-MM and UKESM1-0-LL demonstrate moderate performance.

Among the CMIP6 models analysed, silicate concentrations are generally overestimated across the Southern Ocean (Fig. 5). The three MPI-ESMs exhibit the most significant overestimation, with MBEs exceeding 30 mmol/m³ (Fig. 6), over twice the observed surface silicate concentration of 12.65 mmol/m³ from WOA. The CMCC-ESM2, NorESM2-LM, NorESM2-MM, and UKESM1-0-LL models also show large positive biases, with their mean silicate concentrations roughly double that of observed values (Fig. 6). The CMCC-ESM2 and UKESM1-0-LL models underestimate silicate concentrations in the subtropical zone (STZ), while the two NorESMs underestimate silicate concentrations in the Ross Sea, Weddell Sea, and adjacent waters (Fig. 5). CESM2, CNRM-ESM2-1, GFDL-ESM4, and IPSL-CM6A-LR exhibit the lowest positive MBEs among the models (Fig. 6). and underestimate silicate concentrations in the STZ. Interestingly, in some regions around Antarctica, simulated silicate concentrations are lower than observations, particularly in areas where the GFDL-ESM4 and IPSL-CM6A-LR models overestimate chlorophyll (Fig. 1), suggesting a possible link between silicate availability and diatom growth. Three models, including ACCESS-ESM1-5, CanESM5, and MIROC-ES2L are excluded from the silicate comparison because they do not include diatoms as one of their phytoplankton species or silicate as a nutrient variable.

Figure 5: Observed surface silicate concentrations from WOA in DJF and spatial biases of surface silicate concentrations for 14 CMIP6 models (model silicate – WOA silicate) in DJF for the Southern Ocean (>30°S). Models with unavailable silicate are labelled with \*.

Figure 6: The mean bias errors in surface silicate concentrations for the Southern Ocean (SO), the subtropical zone (STZ), the subantarctic zone (SAZ), the polar front zone (PFZ), and the Antarctic zone (AZ) in DJF. Models with unavailable silicate are labelled with \*.

Among the 11 CMIP6 models with available silicate data, IPSL-CM6A-LR is the best-performing model for representing silicate distribution across the Southern Ocean. It has the lowest MBE (1.50 mmol/m³, compared to the observation of 12.65 mmol/m³), an SSD closest to 1 (1.04), the lowest RMSD (0.37), and the highest CC (0.94; Fig. S9 and Table S4), making it the most reliable model for simulating silicate concentrations. Following IPSL-CM6A-LR, the CNRM-ESM2-1, GFDL-ESM4, and CESM2 models also show relatively good performance, although their statistical metrics are not as strong as IPSL-CM6A-LR. The remaining models, CMCC-ESM2, MPI-ESMs, NorESMs, and UKESM1-0-LL, produce less realistic simulations due to their large bias errors, which suggests significant discrepancies in their silicate simulations.

After evaluating the overall model performance for surface nitrate and silicate, we further examined whether these surface biases are linked to errors in the deep upwelling source waters. To this end, we compared the model-observation biases in nitrate and silicate between the surface and 700 m depth, representing the upwelling Circumpolar Deep Water (CDW) south of  $50^{\circ}$  S. Most models show positive but weak correlations between surface and deep nitrate biases, with slopes ranging from -0.12 to 0.74 and R² from 0.01 to 0.42 (p 

Figure 7: Observed surface dissolved iron concentrations from an integrated product by GEOTRACES IDP2021v2 and Tagliabue et al. (2012), and spatial biases of surface dissolved iron concentrations for 14 CMIP6 models (model dissolved iron – observed dissolved iron) for the Southern Ocean (>30°S). Models with unavailable dissolved iron are labelled with \*.

Figure 8: The mean bias errors in surface dissolved iron concentrations for the Southern Ocean (SO), the subtropical zone (STZ), the subantarctic zone (SAZ), the polar front zone (PFZ), and the Antarctic zone (AZ) in DJF. Models with unavailable dissolved iron are labelled with \*.

Although several models exhibit reasonable MBEs, most perform poorly in other statistical metrics for dissolved iron, with SSD values below 0.51, RMSD exceeding 0.99, and CC lower than 0.18 (Fig. S12 and Table S5). Some models, including the three MPI-ESMs and NorESM2-LM, even show negative CC values, indicating an unrealistic spatial distribution of dissolved iron. Among the models, IPSL-CM6A-LR and CNRM-ESM2-1 perform best when all four statistical metrics (SSD, RMSD and CC) are considered together (Table S5), largely benefiting from their shared ocean biogeochemical module, PISCES-v2 (Aumont et al., 2015). Despite these results, the large overall biases and weak correlations highlight persistent uncertainties in evaluating dissolved iron simulations, primarily due to the limited spatial and temporal coverage of observational data and the oversimplified representation of iron cycling processes in many models. Consequently, it remains difficult to draw definitive conclusions regarding model skill in reproducing dissolved iron distributions in the Southern Ocean.

#### 330 3.2 Performance of DCMs

The observational data from Copernicus indicate that DCMs are widespread across approximately 85% of the Southern Ocean in austral summer (Fig. 9). Their occurrence frequency is lower in the SAZ (below 70%) but exceeds 90% in other regions. Areas without DCMs are primarily located south of Australia, southwest of Chile, and in the Weddell and Ross Seas and surrounding waters. CMIP6 models exhibit varying performance in simulating DCMs. GFDL-ESM4 has DCM occurrence frequency close to 100% across the Southern Ocean (Fig. 10), while the CanESM5 model simulates a DCM frequency similar to observations, but its spatial distribution deviates from observations where we find no DCMs in the Antarctic waters. CNRM-ESM2-1 simulates a high occurrence of DCMs in the STZ and AZ, but a low occurrence in the SAZ and PFZ (Fig. 9). CMCC-ESM2, IPSL-CM6A-LR, and UKESM1-0-LL models simulate DCMs in the STZ but fail to capture them south of the subtropical front (Fig. 9). The ACCESS-ESM1-5, CESM2, MIROC-ES2L, and the three MPI-ESMs models sporadically simulate DCMs in the STZ, resulting in a low overall DCM frequency (

Figure 9: Chlorophyll concentration at deep chlorophyll maximum (DCM) depth during DJF for Copernicus product and 14 CMIP6 models in the Southern Ocean (>30°S). The colours in the maps indicate the chlorophyll concentration at DCM depth, while white areas represent regions where no DCM occurred.

Figure 10: The percentage of DCM occurrence in the Southern Ocean (SO), the subtropical zone (STZ), the subantarctic zone (SAZ), the polar front zone (PFZ), and the Antarctic zone (AZ) in DJF. All percentages in each region are calculated using area-weighted averages.

# 3.3 Particulate organic carbon (POC)

Observed particulate organic carbon (POC) concentrations in the Southern Ocean are higher in Antarctic coastal waters and lower at low latitudes, with elevated concentrations in the Antarctic Circumpolar Current (ACC) regions within the polar front zone (Fig. 11). Model simulations diverge markedly from this pattern. CMCC-ESM2, MPI-ESM-1-2-HAM, and MPI-ESM-1-2-HR generally overestimate POC across most of the basin, except for underestimations in the subtropical zone, yielding MBEs of 18.3, 13.2, and 58.6 mg/m³ (Fig. 12), respectively, compared to the observed mean of 84.1 mg/m³. The overestimated POC concentrations in MPI-ESM-1-2-HAM and MPI-ESM-1-2-HR correspond to their significantly high simulated chlorophyll concentrations (Fig. 1). In contrast, ACCESS-ESM1-5, CanESM5, CESM2, CNRM-ESM2-1, GFDL-ESM4, IPSL-CM6A-LR, and MIROC-ES2L tend to underestimate surface POC across most of the basin. MPI-ESM1-2-LR, two NorESMs, and UKESM1-0-LL slightly overestimate POC in the subtropical zone but also overestimate it south of the subtropical front. ACCESS-ESM1-5, CanESM5, CESM2, and MIROC-ES2L exhibit the largest negative MBEs (exceeding 50 mg m³), severely underrepresenting POC. These strong underestimations are primarily attributable to the absence of a diatom or silicon module in ACCESS-ESM1-5, CanESM5, and MIROC-ES2L and the lack of detrital organic matter output in CESM2 and MIROC-ES2L. Moreover, most models fail to reproduce the elevated POC concentrations observed in the polar front zone, which are sustained by nutrient supply by upwelling and cross-shelf transport, suggesting that the simulated ACC strength in these models may be weaker than observed.

Figure 11: Observed surface POC concentrations from Copernicus in DJF and spatial biases of surface POC concentrations for 14 CMIP6 models (model POC – Copernicus POC) in DJF for the Southern Ocean (>30°S). POC data in CMIP6 models contain phytoplankton carbon, detrital organic carbon, and bacterial carbon.

Figure 12: The mean bias errors in surface POC concentrations for the Southern Ocean (SO), the subtropical zone (STZ), the subantarctic zone (SAZ), the polar front zone (PFZ), and the Antarctic zone (AZ) in DJF. POC data in CMIP6 models contain phytoplankton carbon, detrital organic carbon, and bacterial carbon.

Among 14 CMIP6 models, GFDL-ESM4 provides the most realistic simulations of POC, with an MBE of -24.16 mg m<sup>-3</sup>, an SSD (0.71) close to 1, the smallest RMSD (0.70), and one of the highest CC values (0.71), making it the best-performing model for representing POC (Fig. S13 and Table S6). IPSL-CM6A-LR, UKESM1-0-LL, and CMCC-ESM2 also show strong statistical performance across all metrics. CESM2 performs moderately well due to its favourable SSD, RMSD, and CC values despite exhibiting a relatively large bias error. In contrast, MPI-ESM1-2-LR performs poorly because of its weak statistical metrics, despite having the smallest |MBE|. Other models, including ACCESS-ESM1-5, CanESM5, CNRM-ESM2-1, MIROC-ES2L, MPI-ESM-1-2-HAM, MPI-ESM1-2-HR, and the two NorESMs, are less reliable for simulating POC due to their large biases and poor overall statistical performance.

#### 3.4 Model ranking

Based on the statistical evaluation of surface chlorophyll, nitrate, silicate, dissolved iron and POC using MBE, SSD, RMSD, and CC (Section 3.1 and 3.3), along with DCM occurrence frequency (Section 3.2), we computed a variable-specific and an overall ranking for each model following the methodology described in Section 2.4. The results are shown in Fig. 13 as a heat map. IPSL-CM6A-LR ranks the highest overall, placing within the top two models for all variables. GFDL-ESM4 follows closely, achieving top two rankings across all variables except silicate and dissolved iron, where it ranks third and fourth, respectively. UKESM1-0-LL ranks third, supported by its relatively balanced performance across all metrics. CNRM-ESM2-1, which also incorporates the PISCES-v2 biogeochemical model (as in IPSL-CM6A-LR) ranks third, with performance slightly below that of IPSL-CM6A-LR across most variables. CMCC-ESM2 demonstrates strong performance in chlorophyll, DCM and POC (all rank in the top five), but its lower scores for nutrient variables reduce its overall ranking to fifth. Models such as MIROC-ES2L, CESM2, NorESM2-LM, and NorESM2-MM show moderate performance, ranking from sixth to ninth. CanESM5 and ACCESS-ESM1-5 perform poorly in biogeochemistry due to the absence of key variables (e.g. silicate and dissolved iron), ranking tenth and eleventh. The three MPI-ESMs, all coupled with HAMOCC6, occupy the lowest three positions, with weak performance across all variables. In summary, IPSL-CM6A-LR and GFDL-ESM4 emerge as the most robust models for simulating biogeochemical processes in the Southern Ocean, with consistent and reliable performance across a suite of key biogeochemical indicators.

Figure 13: Heat-map of performance ranks for 12 CMIP6 models. Columns list the evaluated variables—surface chlorophyll (Chl), nitrate (NO3), silicate (Si), dissolved iron (dFe), particulate organic carbon (POC), deep chlorophyll maximum metrics (DCM)—and an overall score (OVR). Rows list the models. Box colours and overlaid numbers give the rank for each modelvariable pair (1 = best, higher numbers = poorer performance): reds indicate higher ranks, blues lower ranks, and grey boxes indicate variables not available for that model.

# 4 Discussion

#### 4.1 Vertical structure of carbon

Most CMIP6 models perform relatively well in simulating surface chlorophyll in the Southern Ocean, but they exhibit only moderate skill in representing surface particulate organic carbon (POC). In contrast, the majority of models struggle to

accurately simulate the deep chlorophyll maxima (DCMs), which are crucial for capturing the vertical structure of chlorophyll distributions. As discussed in Sect 3.2, models such as CanESM5, CNRM-ESM2-1, and GFDL-ESM4 reproduce the horizontal frequency patterns of DCMs reasonably well. However, when surface chlorophyll performance is also considered, GFDL-ESM4 emerges as the only model that satisfactorily represents both surface chlorophyll concentrations and DCM frequency. This finding suggests that most CMIP6 models face challenges in simulating the vertical structure of chlorophyll, as well as POC distributions.

To compare the vertical structure of chlorophyll and POC between models and observations, we integrated their concentrations over the top 100m of the water column, where the majority of primary production occurs (Henley et al., 2020; Arrigo et al., 2008). Unlike the surface chlorophyll and POC, which are generally close to observations, the vertically integrated chlorophyll and POC in the upper 100m are significantly underestimated by most CMIP6 models, except chlorophyll in MPI-ESM-1-2-HR and POC in CMCC-ESM2, both of which are overestimated (Fig. 14).

425

Figure 14: Mean vertical profiles of chlorophyll during DJF (December–January–February) across the Southern Ocean (SO) and its subregions: the Subtropical Zone (STZ), Subantarctic Zone (SAZ), Polar Frontal Zone (PFZ), and Antarctic Zone (AZ), based on observations (Copernicus) and 14 CMIP6 models. Solid lines represent chlorophyll profiles in different regions, while dashed lines indicate the threshold depth of chlorophyll, defined as the depth at which chlorophyll concentration reaches 10% of the maximum value.

The underestimation of vertically integrated chlorophyll in the top 100 m ranges from -63% for CESM2 to -16% for GFDL-ESM4 (Fig. 14) and is influenced by both surface chlorophyll concentrations and the vertical structure of the water column. For example, ACCESS-ESM1-5, CESM2, NorESM2-LM, and NorESM2-MM exhibit similar vertical chlorophyll profiles, characterised by low surface concentrations, almost no deep chlorophyll maxima (DCMs), and shallow chlorophyll threshold depth (CTD; defined as the depth where chlorophyll falls to 10% of the maximum), resulting in underestimations exceeding 50% (Fig. 14). In contrast, MPI-ESM-1-2-HAM and MPI-ESM1-2-LR show high surface chlorophyll levels but extremely shallow CTD (<50 m), leading to low vertically integrated chlorophyll. A third pattern is found in CanESM5, CMCC-ESM2, CNRM-ESM2-1, IPSL-CM6A-LR, MIROC-ES2L, and UKESM1-0-LL, which simulate appropriate threshold depths (~150 m) and some occurrence of DCMs, but their low surface chlorophyll leads to insufficient primary production in the water column. GFDL-ESM4 demonstrates a vertical structure most similar to observations, with a slightly shallower threshold depth, resulting in only an 18% underestimation of integrated chlorophyll. While CMIP6 models vary widely in their simulation of surface chlorophyll concentrations and generally manage to control these levels, they largely lack the capability to accurately simulate the vertical structure of chlorophyll, including both DCMs and CTD.

The vertical structure of chlorophyll and the formation of DCMs are influenced by various environmental and biological factors. Observational evidence indicates that roughly half of DCMs are driven by photoacclimation (Cornec et al., 2021), reflected by the decline in the carbon to chlorophyll (C:Chl) ratio from values exceeding 100 g:g at the surface to below 50 at the base of the euphotic zone (Marañón et al., 2021; Boyd et al., 2024), while the other half are deep biomass maxima (DBMs) driven by the accumulation of phytoplankton biomass below the surface (Cornec et al., 2021). The poor representation of DCMs in ACCESS-ESM1-5 (with its coupled biogeochemical component WOMBAT), the MPI-ESMs (coupled with HAMOCC6), and the NorESMs (coupled with HAMOCC) is therefore likely due to their use of a fixed C:Chl ratio (Oke et al., 2013; Ilyina et al., 2013; Tjiputra et al., 2020), which prevents the simulation of photoacclimation processes. Moreover, most models represent chlorophyll as the biomass of living phytoplankton only, excluding pigments associated with detrital cells that can still be detected in the real water column (Behrenfeld and Boss, 2006). This structural difference contributes to weaker or shallower modelled DCMs compared to observations. In addition, Boyd et al. (2024) suggested that the formation and persistence of DCMs and DBMs can also result from subsurface recycled iron and the ammonium maxima, as well as upward silicate transport that supports diatom production, processes that are often poorly represented in models.

Phytoplankton functional types (PFTs) significantly influence the vertical distribution of chlorophyll. For instance, siliceous diatoms, which account for approximately 75% of primary production in the Southern Ocean (Crosta et al., 2005), are not represented in ACCESS-ESM1-5 and CanESM5. This omission leads to the underestimation of chlorophyll, particularly in the Antarctic zone (Fig. 14). CMIP6 models represent no more than three PFTs, typically small phytoplankton, diatoms, and diazotrophs. In contrast, observational studies, such as Yingling et al. (2025), identify at least five ecologically significant PFTs in the Southern Ocean, including *Synechococcus*, *Picoeukaryotes*, nanoplankton, diatoms, and microplankton. This

simplification of PFT diversity in CMIP6 models likely contributes to inaccurate chlorophyll estimates and unrealistic vertical chlorophyll structures.

The vertical structure of chlorophyll is linked to the mixed layer depth (MLD), which modulates nutrient supply (Durán-Campos et al., 2019; Zampollo et al., 2023). Our analysis indicates a positive correlation between the CTD and MLD (Fig. S14a; R<sup>2</sup>=0.24, p=0.075), suggesting that deep mixing enables phytoplankton to extend further into the water column while maintaining detectable concentrations (Mignot et al., 2014). Conversely, the integrated chlorophyll within the upper 100m shows a negative correlation with MLD (Fig. S14b; R<sup>2</sup>=0.23, p=0.082), likely due to reduced light availability and dilution effects associated with deeper mixed layers (Behrenfeld and Boss, 2006).

Furthermore, the occurrence frequency of DCMs exhibits a Gaussian-like relationship with MLD (Fig. S14c; R<sup>2</sup>=0.42), peaking at MLD of 31 m. When the MLD is excessively shallow, nutrient replenishment to the euphotic zone is limited, inhibiting phytoplankton growth below the surface, thereby reducing the likelihood of DCM formation (Letelier et al., 2004). Conversely, when the MLD becomes too deep, light availability at depth decreases to levels insufficient for sustaining phytoplankton biomass accumulation, which similarly suppresses DCM development (Mignot et al., 2014). Thus, the observed distribution reflects a balance between light limitation from above and nutrient supply from below, a mechanism well-documented in earlier studies (Cullen, 1982; Fennel and Boss, 2003).

# 4.2 Model components and their performance

The performance of CMIP6 models in simulating key biogeochemical variables such as chlorophyll, nitrate, silicate, dissolved iron, POC and DCMs is jointly determined by the complexity of the biogeochemical (BGC) module, the adopted parameterisations of key biogeochemical processes, and the resolution of their coupled ocean and atmosphere model.

Among these, the complexity of the BGC module is the most crucial factor. Key aspects include the representation of phytoplankton functional types (PFTs), stoichiometry flexibility, and nutrient uptake and regeneration schemes. Models that incorporate multiple PFTs, particularly those distinguishing between diatoms and non-diatom phytoplankton, tend to outperform models with a single phytoplankton type in simulating chlorophyll and overall biogeochemical patterns (Fig. 15a; p

Figure 15: Panels show statistical relationships between model rankings and key biogeochemical descriptors: (a) surface chlorophyll ranking vs. inclusion of diazotroph; (c) DCM frequency ranking vs. use of a variable C:Chl ratio; (d) POC ranking vs. presentation of silica cycling (presence of an explicit Si pool or variable C:Si ratio); (e) silicate ranking vs silicate half-saturation coefficient ( $K_{mS0}$ ); (f) nitrate ranking vs. nitrate half-saturation coefficient ( $K_{mN03}$ ); (g) dissolved iron ranking vs. iron half-saturation coefficient ( $K_{mFe}$ ); (h) dissolved iron ranking vs. iron chemistry complexity (simple-no ligand, simple ligand, or complex ligand scheme); (i) DCM frequency ranking vs model ability to assimilate ammonium for photosynthesis. (a), (b), (c), (i) are performed using T-test, (d) and (h) are performed using ANOVA (Analysis of Variance), (e), (f), (g) are performed using linear regression. Tests applied: two-sample t-tests for (a), (b), (c), (i); one-way ANOVA for (d), (h); linear regression for (e)–(g). Each point (colour/shape) represents a CMIP6 model, and dashed lines indicate regression fits where relevant. Corresponding P-values and  $R^2$  statistics (for regressions) are displayed on each panel.

Cellular plasticity (stoichiometry) plays a vital role in regulating nutrient uptake and the cellular elemental composition under variable environmental conditions. Most models employ fixed carbon:nitrogen:phosphorus (C:N:P) ratios consistent with the Redfield Ratio, while carbon:iron ratios are generally dynamic. However, carbon:chlorophyll and carbon:silicate

ratios vary across models. A dynamic carbon:chlorophyll ratio significantly improves the simulation of DCM (Fig. 15c; p<0.01), as mentioned in Section 4.1, while a variable carbon:silicate ratio enhances POC representation (Fig. 15d; p<0.01), especially given the dominance of diatoms Southern Ocean primary production (Crosta et al., 2005).







Phytoplankton growth in models is typically limited by light and nutrient availability, often represented using Michaelis-Menten kinetics (Michaelis and Menten, 1913). However, our analysis did not reveal a clear relationship between model performance in simulating surface chlorophyll or DCMs and specific light or nutrient uptake parameters, such as initial PI (production-irradiance) slope or half-saturation coefficients for nitrate, silicate, and dissolved iron. This suggests that chlorophyll distribution is governed by a complex interplay of environmental drivers rather than any single parameter. In contrast, nutrient concentrations are more directly influenced by process parameterisation. For example, higher silicate halfsaturation coefficients (e.g. 8 mmol/m<sup>3</sup> in PISCES-v2, as used in CNRM-ESM2-1 and IPS-CM6A-LR) spear to improve silicate simulations (Fig. 15e; R<sup>2</sup>=0.52, p=0.01; Nelson et al., 2001). Similarly, nitrate half-saturation coefficients in the range of 1-3 mmol m<sup>-3</sup> tend to yield better agreement with observations (Fig. 15f; R<sup>2</sup>=0.36, p=0.02; Epply et al., 1969). For dissolved iron, no clear correlation was found between model performance and the half-saturation coefficient (Fig. 15g; R<sup>2</sup>=0.01, p=0.81). The complexity of the iron cycle contributes to variability in simulated dissolved iron performance (Fig. 15h; p=0.03). Models with more advanced iron chemistry, such as PISCES-v2 (BGC model coupled in CNRM-ESM2-1 and IPSL-CM6A-LR), which includes strong and weak ligands, and five iron forms (free Fe(II), Fe(III) bounded to strong and weak ligands, and particulate iron) tend to simulate dissolved iron more accurately than those with simple iron complexation (Tagliabue et al., 2023). In contrast, models with simple iron complexation schemes do not show strong ability to simulate better iron concentrations than a simple iron model, which only contains basic iron processes such as scavenging. These inconsistencies are likely due to the limited spatial and temporal coverage of iron observations, which hinders robust evaluation and may mask the benefits of advanced iron cycling mechanisms. Additionally, the utilisation of ammonium appears to promote the formation of DCMs (Fig. 15i; p<0.01), as ammonium-primarily produced through remineralisation-is more readily and rapidly assimilated by phytoplankton than nitrate. This is due to its lower energy and electron requirements for incorporation into cellular biomass. Consequently, substantial ammonium production by heterotrophic bacteria in the subsurface can enhance phytoplankton growth and contribute to the development of DCMs (Boyd et al., 2024).

We also found that the resolution of the ocean component in ESMs can influence the performance of simulated biogeochemical variables. For example, MPI-ESM1-2-HR and MPI-ESM1-2-LR, both coupled with the same biogeochemical model (HAMOCC6), differ significantly in ocean resolution 0.4° vs 1.5°, respectively, and show notable differences in biogeochemical performance. The mean surface chlorophyll concentration in austral summer is 2.37 mg m<sup>-3</sup> in MPI-ESM1-2-HR, compared to 1.35 mg m<sup>-3</sup> in MPI-ESM1-2-LR which is closer to the Copernicus chlorophyll dataset. These discrepancies may arise from resolution-induced differences in ocean circulation and physical conditions, which influence nutrient availability, light penetration, and phytoplankton dynamics. In contrast, variations in atmospheric model resolution appear to have a limited impact on ocean biogeochemistry. For instance, NorESM2-MM and NorESM2-LM,

which use the same ocean biogeochemical model (HAMOCC) but differ in atmospheric resolution (2° vs 1°), exhibit nearly identical biogeochemical outcomes such as mean austral summer surface chlorophyll concentrations of 0.56 and 0.55 mg m<sup>-3</sup>, respectively. These findings suggest that while higher ocean resolution can improve the realism of physical processes affecting biogeochemical simulations, it does not necessarily guarantee better biogeochemical performance.

# 4.3 Avenues for improvement in biogeochemical representation







This study provides a comparative assessment of several ocean biogeochemical indicators for 14 CMIP6 ESMs over the Southern Ocean. Although some models performed adequately, there remain several key directions for future improvements:

- Improvements in the underlying physical ocean models are equally critical for advancing BGC performance. Many biases originate from deficiencies in simulating stratification, mixed layer depth, and large-scale circulation. In particular, key processes such as the entrainment of nutrient-enriched shelf waters along the Patagonian Shelf and the nutrient supply from icebergs and glacial melt along East Antarctica are poorly resolved in coarse-resolution models. Increasing model resolution, refining submesoscale and vertical mixing parameterisations, and enhancing the coupling with sea-ice dynamics and meltwater fluxes will be essential to better capture nutrient transport pathways and the resulting spatial distribution of phytoplankton.
- The representation of key biogeochemical processes in most BGC models remains simplified or parameterised based on limited observations. For instance, differences in the phytoplankton functional types (PFTs), elemental composition (fixed or variable stoichiometry), and nutrient uptake parameterisation contribute to model divergence. Future models should incorporate a more complex marine food web, and more dynamic parameterisations informed by field and laboratory experiments, especially under Southern Ocean specific conditions.
- As the key factor controlling the Southern Ocean primary production, iron cycles and their representations remain poor in most models, compared to limited iron sampled data. Improvements in the simulation of iron sources (e.g., dust deposition, sediment resuspension), bioavailability (i.e., more complex iron chemistry module (Tagliabue et al., 2023), such as including iron-binding ligands), and biological recycling are essential to help reduce the bias in simulated chlorophyll.
- Most models lack a good representation of the vertical structure of chlorophyll and biomass. For example, some models exhibit discrepancies in simulating mixed layer depth and other physical properties, which in turn affects nutrient supply. There is also an oversimplified remineralisation by heterotrophic bacteria, and a lack of diversity of PFTs. Future efforts could expand the model structure to capture these ecological dynamics, which are particularly important in determining vertical profiles and export efficiency for biomass.
- Observational constraints remain limited, especially for subsurface variables such as DCMs, dissolved iron, and POC. Future work should prioritise the integration of additional in situ datasets to validate and improve model

parameterisations. Ensemble data assimilation or machine learning approaches could also be explored for model tuning.

# 570 5 Conclusion





This study evaluated the performance of key biogeochemical variables, including austral summer surface chlorophyll and deep chlorophyll maxima (DCMs), nitrate, silicate, dissolved iron, and particulate organic carbon (POC) across 14 CMIP6 models in the Southern Ocean (south of 30°S). The results reveal substantial variability in model skill. While some models demonstrated strong performance, others showed significant over- or underestimations. Among them, GFDL-ESM4 was the most effective in reproducing surface chlorophyll and POC and DCM features, while IPSL-CM6A-LR performed best in simulating nutrient distribution, such as nitrate, silicate, and dissolved iron. Based on aggregated performance across all variables, the top five models for simulating Southern Ocean biogeochemistry are IPSL-CM6A-LR, GFDL-ESM4, CNRM-ESM2-1, UKESM1-0-LL, and CMCC-ESM2. Our analysis highlights common limitations across CMIP6 models: the underrepresentation of vertical biogeochemical structures, such as the DCM distributions, and the inadequate simulation of physical nutrient transport processes, including upwelling and terrestrial nutrient inputs in productive shelf regions such as the Tasman Sea and the Patagonian Shelf. Additionally, spatial mismatches and persistent biases, particularly for dissolved iron and POC, underscore the need for targeted model improvements. Overall, this study not only provides a comprehensive evaluation of model performance for key biogeochemical variables but also offers insights into areas requiring refinement. These insights can guide future model development and support more informed model selection. Enhancing the representation of biogeochemical processes in Earth system models is essential for improving projections of the Southern Ocean's role in the global carbon and nutrient cycles under ongoing climate change.

#### Code availability

All codes for regridding datasets and data analysis are available at <a href="https://github.com/mingcheng7/Evaluation-CMIP6-historical">https://github.com/mingcheng7/Evaluation-CMIP6-historical</a>.

# 590 Data availability

Raw CMIP6 used in this study are available on the Earth System Grid Federation (ESGF) Nodes for the CMIP6 Archive at <a href="https://esgf.github.io/nodes.html">https://esgf.github.io/nodes.html</a> (Cinquini et al., 2014). Copernicus Global Ocean 3D Chlorophyll-a Concentration, Particulate Backscattering coefficient and Particulate Organic Carbon Product can be accessed at <a href="https://doi.org/10.48670/moi-00046">https://doi.org/10.48670/moi-00046</a> (Sauzède et al., 2016). The Aqua-MODIS chlorophyll concentration in the Southern Ocean by Johnson et al. (2013) can be accessed at <a href="https://portal.aodn.org.au/">https://portal.aodn.org.au/</a>. The World Ocean Atlas (WOA) 2018 data can

be accessed at <a href="https://www.ncei.noaa.gov/access/world-ocean-atlas-2018/">https://www.ncei.noaa.gov/access/world-ocean-atlas-2018/</a> (Garcia et al., 2019). The GEOTRACES IDP2021v2 product can be accessed at <a href="https://www.geotraces.org/geotraces-intermediate-data-product-2021/">https://www.geotraces.org/geotraces-intermediate-data-product-2021/</a>. And the global compilation dataset of dissolved iron can be accessed at <a href="https://www.bodc.ac.uk/geotraces/data/historical/">https://www.bodc.ac.uk/geotraces/data/historical/</a> (Tagliabue et al., 2012).

# 600 Author contribution

MC, NM, and MJE contributed to the conceptualisation of the study and participated in writing and revising the manuscript. MC was responsible for data collection, analysis and figure preparation. NM and MJE provided supervision and guidance throughout the project. All authors reviewed and approved the final version of the manuscript.

# **Competing interests**

The authors declare that they have no conflict of interest.

#### Acknowledgements




This research was undertaken with the assistance of resources and services from the National Computational Infrastructure (NCI), which is supported by the Australian Government. We acknowledge the World Climate Research Programme, which, through its Working Group on Coupled Modelling, coordinated and promoted CMIP6. We thank the climate modelling groups for producing and making available their model output, the Earth System Grid Federation (ESGF) for archiving the data and providing access, and the multiple funding agencies that support CMIP6 and ESGF. The GEOTRACES 2021 Intermediate Data Product version 2 (IDP2021v2) represents an international collaboration and is endorsed by the Scientific Committee on Oceanic Research (SCOR). The many researchers and funding agencies responsible for the collection of data and quality control are thanked for their contributions to the IDP2021v2. We also thank two anonymous reviewers for their helpful comments on our manuscript and the associated editor, Peter Landschützer, for handling our manuscript.

# Financial support

MC was supported by an Australian National University PhD scholarship. MC and MJE were supported by the Australian Research Council Special Research Initiative, Australian Centre for Excellence in Antarctic Science (Project Number SR200100008). NM was supported by the Australian Research Council Centre of Excellence for the Weather of the 21st Century (CE230100012).

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
