# Peer review of "Evaluating the performance of CMIP6 models in simulating Southern Ocean biogeochemistry"

_EGUsphere, 2025_

## Author Response (AR1)

We sincerely thank the Anonymous Referee #1 for the thorough and constructive comments on our manuscript. We appreciate the effort taken to highlight both the strengths and the shortcomings of our study, particularly regarding the use of observational data sets and the robustness of the model ranking exercise. Based on the instructions of the editorial support team of Copernicus Publications, we will provide a point-by-point response and outline the revisions we will undertake below before we make revision on the manuscript. Italic font will be used to distinguish our initial replies from the reviewer' comments. Red colour font will be used to distinguish our responses after revision. The line numbers of our final responses with red colours are referred to the manuscript file with track-changes.

Review of "Evaluating the performance of CMIP6 models in simulating Southern Ocean biogeochemistry" by Ming Cheng et al.

Scope of the manuscript, general comments and recommendation

\_\_\_\_\_

The manuscript by Cheng et al. evaluates the performance of the biogeochemical part of CMIP6 models in reproducing Southern Ocean biogeochemical observations. As the Southern Ocean is one of the regions where biogeochemical models diverge most strongly, this is an important subject for a study, especially since biogeochemical models have become quite a bit more complex on average in the transition from CMIP5 to CMIP6 (Seferian et al, .

The evaluation in the manuscript is performed using the typical tools used in that type of study, namely looking at biases, correlation etc. between model output and climatologies of observations, in the end combining the different metrics into an overall ranking of the models. The study is, however, untypical, in that it attempts to judge the models not only against the 'classical' observations, for which good climatologies are available, namely the macronutrients and chlorophyll, but also against observations of the micronutrient iron estimated depths and chlorophyll levels of deep chlorophyll maxima, where those are present, and finally the concentration of POC and even separately the biomasses of zooplankton, detritus and bacteria. Other 'standard' observations, like satellite-based net primary production, dissolved inorganic carbon and total alkalinity are not taken into account.

While I think that the attempt to include new variables into the assessment of biogeochemical models is a progress, the manuscript does not take into account the uncertain state of our knowledge in many of the variables that the authors use. In my view the mauscript is too uncritical of the observational database that they use to compare the models against, and consequently too confident in the ability to judge model outcomes.

Here are my main criticisms concerning this point:

- Firstly, for their iron validation, the authors use the combination of observed bottle data from Tagliabue et al. (2012). This data is (unlike the attribution of this dataset to GEOTRACES, made in the manuscript, which is simply wrong) mostly a compilation of pre-GEOTRACES data of high quality. Since the publication of this data set, a large number of additional data has become available through the GEOTRACES intermediate data products, especially for the Southern Ocean. Why has this data not been taken into account?

We acknowledge the error in attributing the Tagliabue et al. (2012) compilation to GEOTRACES. This will be corrected with the addition of the GEOTRACES IDP2021 reference. In addition, we will extend the iron evaluation to include the most recent GEOTRACES Intermediate Data Product (IDP2021v2) for the Southern Ocean and repeat the comparison. This will improve the robustness of our iron assessment.

We have corrected the dissolved iron comparison with the addition of the GEOTRACES IDP2021v2 product. The new dissolved iron reference data is the merged dataset of GEOTRACES IDP2021v2 and the global compilation in Tagliabue et al. (2012). The method of merging two dataset has been detailly added in the method part in lines 177-188. At the same time, we have updated the figures and contents of the dissolved iron comparison. Additionally, other figures, such as rankings and statistical relationships, have been updated.

- For the evaluation of the depth of the deep chlorophyll maximum and chlorophyll concentration at the maximum, the authors have chosen the product from Copernicus, which is based on the works of Sauzede et al. (2016). The authors mention that this dataset estimates POC and chlorophyll using a neural network method, but do not give any further details. Here is therefore my summary of the method: The data set estimates the vertical distribution of particle backscatter (which can be used as a measure of POC) from the large data base of ARGO vertical profiles of temperature and salinity, and co-located surface satellite estimates of particle backscatter and chlorophyll a from MODIS. Actually, contrary to the statement made in the manuscript, the method presented in Sauzede et al (2016) only describes the estimation of POC profiles, NOT of chlorophyll. For the chlorophyll estimation one should probably cite the data manual (https://documentation.marine.copernicus.eu/QUID/CMEMS-MOB-QUID-015-010.pdf). While this data set is unique in that it for the first time allows a look at the vertical distribution of biological activity in the ocean, it is not 'observations' (which is how it is repeatedly referenced to in the manuscript), but a fairly indirect estimate. The limits of this data set and its possible errors are not discussed at all in this manuscript, and neither are the error estimates, which are present in the data themselves, taken into account in the model assessment. Instead, the data set is uncritically taken as 'truth'.

We agree that the Copernicus product for chlorophyll and POC is an indirect reconstruction based on neural network methods, not direct observations. As mentioned in the user manual of the Copernicus product, the current vertical chlorophyll profile product is generated by the latest neural network-based method called SOCA2024, upgraded from SOCA2016 described in Sauzède et al. (2016). That means, the chlorophyll product is uses the same method as with POC product, even though Sauzède et al. (2016) only mentioned the neural method to produce vertical POC profiles. We will explicitly state that these are observation-based estimates, not direct Chl a measurements. We will also discuss the uncertainties and potential errors in the data set. And we will adjust the language throughout to avoid overstating confidence.

We have added some texts related to the description of the Copernicus chlorophyll and POC dataset in the method part, including the data source, method and some statistics to support the reliability of this product. See lines 120-144.

- Why is the same data set also taken for the evaluation of surface chlorophyll and POC? As the processing of the data in the copernicus product involves chlorophyll and backscatter estimates from MODIS, it would remove one possible source of error to directly use the satellite data here. Actually at this point it should be discussed that the standard algorithm used in satellite estimates of chlorophyll has been questioned in the Southern Ocean by Johnson et al. 2009 (which is cited in the manuscript); the algorithm proposed in Johnson et al. 2009 gives on average higher values of chlorophyll in the Southern Ocean than the standard algorithm used at that time for SeaWIFS. I think this also hold for the GlobColor product used in the copernicus data set, but I have to admit that this is getting beyod my expertise. But I think it illustrates yet another source of uncertainty in the 'observations' that should be discussed.

We acknowledge that using the analysed Copernicus data set for surface chlorophyll and POC is not ideal. In the revised manuscript, we will include direct satellite-based product (MODIS) as an independent comparison for surface chlorophyll and POC. We will also discuss the uncertainty of Southern Ocean chlorophyll satellite-based product as highlighted by Johnson et al. (2013), and how this may affect inter-model comparison and model-observation comparison.

We kept using Copernicus chlorophyll and POC product for comparison in surface chlorophyll and POC after comparing the statistics of Copernicus product, three satellite (SeaWIFS, Aqua-MODIS, and GlobColor) products and the three satellite products refined using the algorithms in Johnson et al. (2013). The three satellite products originally provided by NASA are proven to underestimate chlorophyll (Johnson et al., 2013). The Copernicus product has a stronger ability to estimate chlorophyll (R²=0.89, slope=1.04) and POC (R²=0.93, slope=0.89) when compared to BGC-Argo. And when compared to HPLC chlorophyll data, Copernicus shows R²=0.53 and slope=0.70 (Sauzède et al., 2024). The MODIS data processing using Johnson et al. (2013) algorithm shows an R²=0.51 and slope=0.90. To maintaining consistency across different comparisons (i.e. chlorophyll, DCMs, and POC), we decided to stick with the Copernicus data as it merges both satellite and BGC-Argo data and is proved to be reliable against HPLC pigment data (Sauzède et al., 2024). We also present the chlorophyll comparison using MODIS in Johnson et al. (2013) in the supplement. In the manuscript, we\e have provided related information in the method part in lines 120-144.

- Just out of curiosity: Many model assessments also use satellite-based estimates of net primary production. Is there a specific reason why this was not done here?

We initially did the evaluation of NPP performance. When considering simulation on phytoplankton may be overweighted in model ranking and length of manuscript, we decided not to put NPP evaluation in the manuscript. Of course, we will consider putting the NPP data back to the manuscript.

We finally did not assess the NPP in models. The existing NPP estimating algorithms exhibit large discrepancies (Silsbe et al., 2016) and extensive field measurements of NPP are lacking to constrain the NPP model. Therefore, the current NPP products are not reliable enough to serve as a reference for ranking the models. So, we decided not to include the NPPP assessment in the models. Instead, we present the spatial biases and MBE figures in the supplement information for readers interested in CMIP6 model NPP and this compares to the VGPM model and CAFE model estimates. We have added the related content into the method part of the manuscript and clearly state the reason we did not evaluate NPP in lines 149-157.

- And finally, the authors use ONE number of how POC is distributed over phytoplankton, zooplankton, dead organic matter and bacteria that has been estimated for the Southern Ocean to convert the copernicus estimate of POC into one of phytoplankton, zooplanton, detritus, and bacterial carbon biomass. In their tables 6 and 7 they then judge whether models 'underestimate zooplankton' etc. But when you actually read the paper by Yang et al. 2022, one immediately realizes the limits of that comparison. Firstly, the paper does not describe microzooplankton, but only the biomass of zooplankton that can be caught in plankton nets. Secondly, the biomasses of the three zooplankton groups studied in that paper (mesozooplankton, krill and salps) has a large regional variability, as for example shown in their figure 2. While the Yang paper indeed demonstrates that there is an inverted trophic pyramid in the Southern Ocean, the actual biomass numbers probably have a large uncertainty from sampling bias. Taking the one biomass number for the whole Southern Ocean obtained here then for conversion of a totally different POC estimate into zooplankton biomass further leads to errors. To add to that, the authors do not describe how they have combined the estimates from the three different papers cited into one. In my view it makes sense to investigate whether models obtain a similar inverted trophic pyramid as described in Yang et al, but not to write sentences like 'Most models describe integrated phytoplankton carbon reasonably well with values comparable to observations' when the observations are just indirect estimates of POC from copernicus, multiplied by one Southern Ocean estimate of the phytoplankton carbon:POC ratio, and then not taking possible erors into account. The whole section starting line 412 to line 445 in my view should be scrapped.

We accept that our approach of applying a single partitioning ratio from Yang et al. (2022) is oversimplified and neglects large regional variability and sampling uncertainties. In our original research, we used annual POC data to avoid the effect of missing monthly data of some models. In this case, most models underestimated surface POC concentration according to that the effect of low data availability in winter months on calculation of annual mean. We did some work on classifying carbon type to address the potential points that the types of carbon in the models may have biases. In the revised manuscript, we will redo the surface POC comparison, by changing the annual comparison to summer comparison, to reduce the effect of errors on annual mean calculation. Also, we will delete the section between line 412 and line 445 about discussing POC classification.

We have deleted the section about discussing POC classification. On the one hand, errors exist in Copernicus POC product. On the other hand, using a constant ratio to define the classification of POC in the water column might be inappropriate.

Given these criticisms I don't think the paper can be published without quite major revisions. To make it publishable, I think the following needs to be done:

- Extend the data set used for the comparison of modeled iron by the data from the lates GEOTRACES intermediate data product and repeat the comparison.

Dissolved iron comparison will be repeated using GEOTRACES IDP2021v2.

We have redone the iron comparison using the merged dataset of GEOTRACES IDP2021v2 and the global compilation of measurement in Tagliabue et al. (2012). See lines 346-386.

- Redo the comparison of deep chlorophyll maximum frequency and chlorophyll levels taking into account the uncertainty of the copernicus data set.

The uncertainty of the Copernicus data set will be discussed in comparison of DCM.

We have added the contexts in methods to explain the performance of Copernicus product compared to measured datasets in lines 120-144.

- use (at least in addition to the copernicus data set) the direct satellite-based estimated of chlorophyl and POC from MODIS for the surface comparison; possibly also discuss the issue of the chlorophyll algorithm uncertainty raised by Johnson et al, 2009.

Surface chlorophyll and POC comparison will be repeated using MODIS data set. And the uncertainty of chlorophyll algorithm will be discussed.

We have continued to use the Copernicus chlorophyll and POC product for comparison with the CMIP6 models. We selected the Copernicus product after considering the statistical robustness to measured datasets and those of multi-satellite products in lines 120-144.

- either remove the comparison with the different components of POC completely or do it properly by accounting for the error margins

The comparison with the different components of POC will be completely removed.

We have removed the comparison with different components of POC.

I think all these changes would probably be incompatible with the strong focus of the paper on 'ranking' of the different models, i.e. saying which one is 'the best', which comes second etc. Given the uncertainty of the data sets used, which is completely neglected in the present manuscript, I don't really think this can be done with any confidence.

As this will require more or less a complete rewrite of the manuscript

I limit my further specific comments to the most important ones.

Specific comments
* * *
Line 135-136: '.. we use yearly data instead, as carbon export predominantly occurs during summer months': I don't understand the reasoning here. If carbon export predomnantly occurs in summer, does not using annual POC values make the connection of export less reliable?

As we mentioned above, we will redo the surface POC comparison by changing the annual comparison to a summer comparison. The new POC comparison will include all 14 models.

We have redone the surface POC comparison by changing the annual comparison to an austral summer comparison. Additionally, we have removed zooplankton in the modelled POC as zooplankton is almost not included in the observed POC dataset.

Formula 4: The formula for root-mean-square difference is given here corectly; but in the Taylor diagnam one should use the RSMD after correction for the mean model-data bias, otherwise the connection between CC, SSD and RSMD that is used to construct the diagram does not hold (Taylor 2001). Was this done here?

We ensure that the RMSD values were bias-corrected, so the Taylor diagrams were correctly plotted.

They were initially done before. Additionally, we removed the four basic statistical equations as requested by reviewer 2.

line 153: 'the number of grid points..' Does that depend on the grid resolution? Is that a problem?

Actually, the DCM frequency is calculated based on the area of the grid points, not simply the number of grid points. We will change "the number of grid points" to "the area proportion".

We have changed "the number of grid points" to "the area proportion" in line 202.

Table S1: Were the calculations of CC and other statistical quantities for chlorophyll done using log-transformed data, as is done most of the times?

We did not apply log transformation when calculating CC and other statistical metrics for chlorophyll. For visualisation, we used a moving scale.

Comparison of surface nitrate and silicate: Given that the Southern Ocean is an upwelling region, would it make sense to also check the concentration of these nutrients in Circumpolar Depp water with data when tryng to explain the model-data difference at the surface?

This is a sensible suggestion. In our manuscript, we mainly focus on biogeochemical performance and the effect of biogeochemical processes on biogeochemical performance. We acknowledge that CDW nutrient concentrations influence surface fields in the Southern Ocean, we will consider comparing CDW nutrient concentrations although this analysis is beyond our current scope. We will need to explore this.

We have compared the surface nitrate/silicate to that at 700m in lines 335-345. The nitrate and silicate between the two depths do not show a strong correlation in most models. That is, the surface nutrients may be partially regulated by CDW, and also regulated by biological uptake, vertical mixing, and mixed-layer variability. We have added text discussing this. See lines 335-345.

When comparing dissolved iron with the Tagliabue et al. 2012 data set, mean bias estimates are given. Does a mean make sense in such a sparse data set? Should one perhaps at least also have a look at the median?

The dissolved iron data from Tagliabue et al. (2012) are distributed to  $1^{\circ} \times 1^{\circ}$  grids by calculate their median of closest samples to plot the surface dissolved iron map. In this case, we compared the dissolved iron difference by calculating the mean. We will provide more details on how we used the iron dataset and how we have utilised the GEOTRACES IDP2021v2 data product, noting that most of the Tagliabue et al. (2012) included data from the IPY 2007-2008.

We have provided a detailed method for processing the dissolved iron data in lines 169-180.

In the iron comparison, repeatedly the 'limited availability of observational data' is referred to, which is correct. But the data is not that limited, given the GEOTRACES data that is ignored here.

We will redo the dissolved iron comparison by using the latest GEOTRACES product (IDP2021v2).

We have redone the dissolved iron comparison by using the merged dataset of GEOTRACES IDP2021v2 and Tagliabue et al. (2012). There are finally 615 grid points with available dissolved iron in 1 degree mesh grid, accounting for 4.33% of the total ocean grid points in SO. See lines 169-180.

Model ranking: it is unclear to me how the different statistical quantities to judge model-'data' agreement are converted into one ranking. Is the lowest RSMD the criterium, the highest CC?

The overall ranking of each model is based on its ranking of the different variables. The ranking of a variable for a model is based on rankings of four statistics: MBE (the lowest |MBE| have highest ranking), SSD (the closest to 1 have highest ranking), RMSD (the lowest have highest ranking), and CC (the highest have highest ranking). We will more detailly describe the criterium of ranking in the method section.

We have provided a detailed description of how the model ranking works in lines 203-211.

Line 383: "DCMs are primarili driven by photoacclimation". No, not all of them, see Cornec et al. 2021. The whole discussion of DCMs and the factors driving them is a bit superficial.

We agree with that not all of DCMs are driven by photoacclimation. Cornec et al. (2021) indicated that around half of DCMs are driven by photoacclimation and another half are DBMs. This situation in models is different to conditions within the "real" water column. In models, chlorophyll only represents live phytoplankton, while it will be excluded from the count after phytoplankton dies and is transfer to the detritus pool. However, in the real water column, chlorophyll can also be detected in died phytoplankton. In addition, Boyd et al. (2024) suggested DCM and DBM formation and persistence can be a result from recycled iron within the subsurface associated with the maximum in ammonium and upward silicate transport from depth which support diatom production. The challenge is most models do not simulate this well. These structural difference between the real water column and models makes simulating DCMs in models challenging. In this case, the modelled DCMs are not as strong as them discovered in the water column. We will add related content to the manuscript to interpret the bias on DCMs between observation and simulation.

We changed the expression by stating that roughly half of DCMs are driven by photoacclimation and added some related texts mentioned above. See lines 521-537.

| References |  |  |  |
|------------|--|--|--|
|            |  |  |  |
|            |  |  |  |

Cornec, M., Claustre, H., Mignot, A., Guidi, L., Lacour, L., Poteau, A., et al. (2021). Deep chlorophyll maxima in the global ocean: Occurrences, drivers and characteristics. Global Biogeochemical Cycles, 35, e2020GB006759. https://doi. org/10.1029/2020GB006759

We sincerely thank the Anonymous Referee #2 for the thorough and constructive comments on our manuscript. We appreciate the effort taken to highlight both the strengths and the shortcomings of our study, particularly regarding the use of observational data sets and the robustness of the model ranking exercise. Based on the instructions of the editorial support team of Copernicus Publications, we will provide a point-by-point response and outline the revisions we will undertake below before we make revision on the manuscript. Italic font will be used to distinguish our initial replies from the reviewer' comments. Red colour font will be used to distinguish our responses after revision. The line numbers of our final responses with red colours are referred to the manuscript file with track-changes.

The manuscript "Evaluating the performance of CMIP6 models in simulating Southern Ocean biogeochemistry" analyzes coupled carbon-climate Earth system model fidelity for surface chlorophyll, nitrate, silicate, and iron, the deep chlorophyll maximum, and particular organic carbon across subregions of the Southern Ocean to rank the models which is a highly valuable analysis given the historical challenges in both observations collection and model fidelity and importance of the Southern Ocean for heat and carbon uptake. The biggest weakness of the current manuscript is the assumption that inter-model differences and biases should be attributed to biogeochemical formulation rather than the underlying physics, including representation of temperature, mixed layer depth, upwelling, and upper ocean stratification, transport, and turbulence, all of which are long standing challenges in the climate community. While a detailed discussion of the potential of physical biases and their potential implications is outside the scope of the present manuscript, the possibility of physical attribution should be mentioned. Otherwise I have only minor comments.

We will mention that the underlying physics of each will be mentioned and that this can influence the performance of the BGC component of the model, especially in coastal system.

We have mentioned some physical processes may affect the BGC performance, such the entrainment of iron-rich shelf waters by the iron-poor Antarctic Circumpolar Current through the Drake Passage, and the iron inputs from melting icebergs and sediments along the East Antarctic shelf, which can modulate nutrient and chlorophyll distributions. See lines 221-225, lines 335-345, lines 352-359, and lines 431-433.

Specific comments by line number:

9 - This assertion is highly overstated - see lines 67-82 which contradict this as well as such literature as:

Frölicher, T.L., Sarmiento, J.L., Paynter, D.J., Dunne, J.P., Krasting, J.P. and Winton, M., 2015. Dominance of the Southern Ocean in anthropogenic carbon and heat uptake in CMIP5 models. Journal of Climate, 28(2), pp.862-886.

Mongwe, N.P., Vichi, M. and Monteiro, P.M., 2018. The seasonal cycle of p CO 2 and CO 2 fluxes in the Southern Ocean: diagnosing anomalies in CMIP5 Earth system models. Biogeosciences, 15(9), pp.2851-2872.

Rickard, G.J., Behrens, E., Chiswell, S., Law, C.S. and Pinkerton, M.H., 2023. Biogeochemical and physical assessment of CMIP5 and CMIP6 ocean components for the southwest Pacific Ocean. Journal of Geophysical Research: Biogeosciences, 128(5), p.e2022JG007123.

Nevison, C.D., Manizza, M., Keeling, R.F., Stephens, B.B., Bent, J.D., Dunne, J., Ilyina, T., Long, M., Resplandy, L., Tjiputra, J. and Yukimoto, S., 2016. Evaluating CMIP5 ocean biogeochemistry and Southern Ocean carbon uptake using atmospheric potential oxygen: Present-day performance and future projection. Geophysical Research Letters, 43(5), pp.2077-2085.

We will change open sentence of the abstract "yet the quality of its representation in Earth System Models (ESMs) remains unquantified" to "yet comprehensive assessments of its representation in Earth System Models (ESMs) are still limited"

We have changed the open sentence of the abstract "yet the quality of its representation in Earth System Models (ESMs) remains unquantified" to "yet comprehensive assessments of its representation in Earth System Models (ESMs) are still limited" in lines 9-10.

48 - which? High iron requirement?

More favourable nutrient condition is iron and silicon supply.

We have added "particularly the supply of iron and silicon" in line 49.

60 - by "integration of". do the authors mean "assessment with"? It is not clear what "data" is integrated into these models to represent the Southern Ocean except for topography and radiative forcing.

Models are constrained by some observed dataset. We will change "integration of" to "constrained by".

We have changed "integration of" to "constrained by" in line 61.

99 - Why define the acronym when it is not used again until the acknowledgments and also defined there?

We will remove "(NCI)" at line 99 and line 563.

We have removed "(NCI)" at line 99. For the acronym in acknowledgement, this is the template of acknowledgement for NCI so we do not remove it.

Eq 1-4 - These are all pretty common statistical definitions which could be removed for space.

We will remove these equations.

We have removed these equations.

167 – "MPI-ESM models" should be "MPI-ESMs"

We will change "MPI-ESM models" to "MPI-ESMs" at line 167, 250, 300, 337, and 386.

We have changed "MPI-ESM models" to "MPI-ESMs".

225 - Should be "Fig. 1" to point to chlorophyll.

We will change "Fig. 5" at line 225 to "Fig. 1".

We have changed "Fig. 5" at line 300 to "Fig. 1".

510-558 - The attribution here to biological complexity seems to assume that the Southern Ocean physics that drives the biogeochemistry is perfect in these models. This is not the case and is the subject of many papers. Much of the focus has been on wind and sea ice biases and upper ocean stratification (e.g. Beadling et al, 2020), temperature, (Luo et al., 2023 and polynya (Mohrmann et al., 2021):

Beadling, R. L., Russell, J. L., Stouffer, R. J., Mazloff, M., Talley, L. D., Goodman, P. J., ... & Pandde, A. (2020). Representation of Southern Ocean properties across coupled model intercomparison project generations: CMIP3 to CMIP6. Journal of Climate, 33(15), 6555-6581.

Luo, F., Ying, J., Liu, T., & Chen, D. (2023). Origins of Southern Ocean warm sea surface temperature bias in CMIP6 models. npj Climate and Atmospheric Science, 6(1), 127.

Mohrmann, M., Heuzé, C., & Swart, S. (2021). Southern Ocean polynyas in CMIP6 models. The Cryosphere, 15(9), 4281-4313.

We accept the reviewer's critique that this section over-attributes biases to biological complexity without considering physical drivers. Although a detailed discussion of the potential of physical biases and their potential implications is outside the scope of the manuscript, we will add some content to discuss the effect of physical attributes on biogeochemical performance.

In the case that we have mentioned some physical processes may affect BGC performance, we have added a paragraph to state the future improvement for physical processes in lines 660-666.

**References**

Boyd, P. W., Antoine, D., Baldry, K., Cornec, M., Ellwood, M., Halfter, S., Lacour, L., Latour, P., Strzepek, R. F., Trull, T. W., and Rohr, T.: Controls on Polar Southern Ocean Deep Chlorophyll Maxima: Viewpoints From Multiple Observational Platforms, Global Biogeochemical Cycles, 38, 10.1029/2023gb008033, 2024.

Cornec, M., Claustre, H., Mignot, A., Guidi, L., Lacour, L., Poteau, A., D'Ortenzio, F., Gentili, B., and Schmechtig, C.: Deep Chlorophyll Maxima in the Global Ocean: Occurrences, Drivers and Characteristics, Global Biogeochemical Cycles, 35, 10.1029/2020gb006759, 2021.

Johnson, R., Strutton, P. G., Wright, S. W., Mcminn, A., and Meiners, K. M.: Three improved satellite chlorophyll algorithms for the Southern Ocean, Journal of Geophysical Research: Oceans, 118, 3694-3703, 10.1002/jgrc.20270, 2013.

Sauzède, R., Renosh, P. R., Schemechtig, C., Uitz, J., and Claustre, H.: Quality Information Document for Global Ocean 3D Particulate Organic Carbon and Chlorophyll-a concentration Product MULTIOBS GLO BIO BGC 3D REP 015 010, 2024.

Sauzède, R., Claustre, H., Uitz, J., Jamet, C., Dall'Olmo, G., D'Ortenzio, F., Gentili, B.,

Poteau, A., and Schmechtig, C.: A neural network-based method for merging ocean color and Argo data to extend surface bio-optical properties to depth: Retrieval of the particulate backscattering coefficient, Journal of Geophysical Research: Oceans, 121, 2552-2571, 10.1002/2015jc011408, 2016.

Silsbe, G. M., Behrenfeld, M. J., Halsey, K. H., Milligan, A. J., and Westberry, T. K.: The CAFE model: A net production model for global ocean phytoplankton, Global Biogeochemical Cycles, 30, 1756-1777, 10.1002/2016gb005521, 2016.

Tagliabue, A., Mtshali, T., Aumont, O., Bowie, A. R., Klunder, M. B., Roychoudhury, A. N., and Swart, S.: A global compilation of dissolved iron measurements: focus on distributions and processes in the Southern Ocean, Biogeosciences, 9, 2333-2349, 10.5194/bg-9-2333-2012, 2012.

Yang, G., Atkinson, A., Pakhomov, E. A., Hill, S. L., and Racault, M. F.: Massive circumpolar biomass of Southern Ocean zooplankton: Implications for food web structure, carbon export, and marine spatial planning, Limnology and Oceanography, 67, 2516-2530, 10.1002/lno.12219, 2022.